# Glitazone Treatment Rescues Phenotypic Deficits in a Fly Model of Gaucher/Parkinson’s Disease

**DOI:** 10.3390/ijms222312740

**Published:** 2021-11-25

**Authors:** Oluwanifemi Shola-Dare, Shelby Bailess, Carlos C. Flores, William M. Vanderheyden, Jason R. Gerstner

**Affiliations:** 1Elson S. Floyd College of Medicine, Washington State University, Spokane, WA 99202, USA; o.shola-dare@wsu.edu (O.S.-D.); shelby.bailess@wsu.edu (S.B.); carlos.c.flores@wsu.edu (C.C.F.); w.vanderheyden@wsu.edu (W.M.V.); 2Steve Gleason Institute for Neuroscience, Washington State University, Spokane, WA 99202, USA

**Keywords:** neurodegeneration, autophagy, Lewy body, dementia, p62, β-acid glucosidase 1

## Abstract

Parkinson’s Disease (PD) is the most common movement disorder, and the strongest genetic risk factor for PD is mutations in the glucocerebrosidase gene (*GBA*). Mutations in *GBA* also lead to the development of Gaucher Disease (GD), the most common type of lysosomal storage disorder. Current therapeutic approaches fail to address neurological GD symptoms. Therefore, identifying therapeutic strategies that improve the phenotypic traits associated with GD/PD in animal models may provide an opportunity for treating neurological manifestations of GD/PD. Thiazolidinediones (TZDs, also called glitazones) are a class of compounds targeted for the treatment of type 2 diabetes, and have also shown promise for the treatment of neurodegenerative disease, including PD. Here, we tested the efficacy of glitazone administration during development in a fly GD model with deletions in the *GBA* homolog, *dGBA1b (GBA1*^ΔTT/ΔTT^). We observed an optimal dose of pioglitazone (PGZ) at a concentration of 1 μM that reduced sleep deficits, locomotor impairments, climbing defects, and restoration of normal protein levels of Ref(2)P, a marker of autophagic flux, in *GBA1*^ΔTT/ΔTT^ mutant flies, compared to *GBA1*^+/+^ control flies. These data suggest that PGZ may represent a potential compound with which to treat GD/PD by improving function of lysosomal-autophagy pathways, a cellular process that removes misfolded or aggregated proteins.

## 1. Introduction

Parkinson’s Disease (PD) was first described by James Parkinson, in 1817, as the Shaking Palsy, and is the second most common age-related neurodegenerative disorder after Alzheimer’s Disease. PD may be genetic (familial) or non-genetic (sporadic). Only 10–20% of PD cases are familial PD, and there have been 21 loci identified in familial PD. Mutations in several genes, including *SNCA*, *LRRK2*, *MAPT*, and *GBA1*, have been identified as leaving carriers susceptible to PD. Genetic risk factors also affect age of onset and the disease progression [1]. The most common and strongest genetic risk factor for PD is mutations in *GBA1* [2,3,4,5]. *GBA1* encodes the lysosomal enzyme β-acid glucosidase 1 or glucocerebrosidase. The GBA1 enzyme is a 497 amino acid membrane-associated protein synthesized in the endoplasmic reticulum in an inactive state; however, it becomes active when transferred to the acidic lumen of the lysosome. Once transported to the lysosome, GBA1 interacts with its activator protein, saposin C (SAPC), to catalyze the hydrolytic degradation of glucose moieties, such as glucosylceramide to ceramide and glucose, and glucosylsphingosine to glucose and sphingosine [2,4]. Elevated levels of glucosylceramides have also been observed in the brains of patients with PD without *GBA1* mutations [6,7], suggesting that the lysosome autophagy pathway may be disrupted in both sporadic and familial PD cases.

Mutations in *GBA1* lead to low or deficient levels of the GBA1 enzyme, also resulting in the development of the recessively inherited lysosomal storage disorder known as Gaucher Disease (GD). In patients with GD, the breakdown of glucosylceramide is insufficient, leading to the accumulation of substrates (glucosylceramide) within the lysosome. There are at least 495 known *GBA1* mutations that have been found to be associated with GD. Much like PD, GD is highly heterogenous and has been categorized into three types based on clinical manifestations. Mild mutations, such as N370S, are associated with type 1, the non-neuronopathic form, while severe mutations, such as L444P, which result in mutant enzyme production, are associated with type 2, the acute neuronopathic form, or type 3, the subacute neuronopathic form. Both type 2 and type 3 are characterized by neurological impairment. Recent studies suggest that the L444P mutation worsens cognitive decline and the overall progression of PD. Clinical manifestations of GD-PD are almost identical to those of sporadic PD. Notably, the risk of progression to dementia for patients with GD-associated PD is approximately five times that of sporadic PD. Additionally, patients with GD-associated PD exhibit a fourfold increase in the risk of progressing to a stage with impaired quality of life [8,9].

The common fruit fly, *Drosophila melanogaster*, serves as a useful model organism for the study of human neurodegenerative diseases [10] because it shares approximately 75% of human disease-related genes [11], and it is an ideal organism for screening small molecules for drug discovery [12]. *Drosophila* possess two *GBA1* homologs, *CG31148* (*dGBA1a*) and *CG31414* (*dGBA1b*). Each of these genes share 50% similarity with human *GBA*. The homolog *dGBA1a* is primarily expressed in the midgut, while *dGBA1b* is ubiquitously expressed in a wide range of tissues, including the larval and adult brain [13,14]. Several GBA-PD fly models have been developed to better understand the relationship between GBA mutations and PD pathophysiology [15,16,17,18]. The neuronopathic GBA deficiency fly model (*GBA1*^ΔTT/ΔTT^) features a truncated version of expressed *dGBA1a* and a deletion in *GBA1b*, leading to a 60% reduction in GBA1 enzyme activity in fly heads, compared to control *GBA1^+/+^* flies, along with reduced lifespan and age-dependent behavioral abnormalities, including climbing defects [15]. Additionally, *GBA1*^ΔTT/ΔTT^ homozygote flies feature a decreased ability to degrade ubiquitinated proteins via autophagy [15].

Thiazolidinediones (TZDs) include pioglitazone, troglitazone, and rosiglitazone [19,20,21,22]. Several studies have shown that pioglitazone and rosiglitazone exert neuroprotective and anti-inflammatory effects in models of PD; it is believed that these effects are exerted through peroxisome proliferator-activated receptor γ (PPARγ) coactivator-1α (PGC-1α) activation [23,24,25,26,27]. Administration of pioglitazone in an MPTP rhesus monkey model of PD showed a reduction in dopaminergic neurodegeneration and the infiltration of CD68-positive macrophages in the nigrostriatal area [28]. Several studies have shown that TZD treatment in diabetic patients is associated with a reduction in incidence of PD [29,30,31]. A recent study examined the effects of PGZ treatment specifically in patients with diabetes and found that there was a dose-dependent reduction in the incidence of PD [32].

In this study, we were interested in determining whether TZD drugs can improve the phenotypic deficits associated with PD/GD in a fly model. We found that raising flies throughout development until adulthood in the presence 1 μM pioglitazone (PGZ) was able to reduce sleep deficits, locomotor impairments, and climbing defects in young male *GBA1*^ΔTT/ΔTT^ mutant flies, compared to male *GBA1*^+/+^ control flies. PGZ treatment at this dose also restored protein levels of Ref(2)P, the fly homolog of p62 and marker of autophagic flux, in *GBA1*^ΔTT/ΔTT^ mutant flies to control levels in *GBA1*^+/+^ flies. These results suggest that PGZ may offer therapeutic potential in the treatment of GD/PD by restoring the normal function of lysosomal autophagy pathways.

## 2. Results

### 2.1. Sleep-Wake Behavior

#### 2.1.1. Changes in Sleep-Wake Behavior in GBA1^ΔTT/ΔTT^ Mutants

Fruit flies exhibit a circadian rest-activity cycle that shares many characteristics with mammalian sleep, including rodents and humans [33]. Sleep disturbances are one of the most common and disabling nonmotor manifestations of PD, affecting up to 90% of patients with PD [34]. To determine the consequences of glucocerebrosidase deficiency on sleep-wake behavior in our GD fly model, we measured the rest-activity of individual GBA1^ΔTT/ΔTT^ mutant and GBA1^+/+^ control flies using the Drosophila Activity Monitoring system, DAMS [35,36]. We observed significant reductions in total sleep time, daytime sleep, and nighttime sleep in the GBA1^ΔTT/ΔTT^ mutants compared to the control GBA1^+/+^ flies (Appendix A Figure A1).

#### 2.1.2. Thiazolidinedione (Glitazones) Screening in the GBA1 Deficiency Fly Model of PD

To assess the neuroprotective effects of glitazones in our GBA deficiency fly model of PD, we used the sleep deficit phenotype to screen for potential neuroprotective effects of the glitazone drugs troglitazone (TGZ) and pioglitazone (PGZ) at three different doses: 500 nM, 1 μM, and 2 μM. Raising flies in the presence of TGZ failed to reduce daytime sleep deficits in GBA^ΔTT/ΔTT^ mutants compared to control GBA1^+/+^ flies at 500 nM, 1 μM and 2 μM doses (Figure 1A–C). Treatment with 500 nM TGZ significantly reduced nighttime sleep deficits in the GBA^ΔTT/ΔTT^ mutants (Figure 1D) but failed to have any effects at higher 1 μM and 2 μM doses (Figure 1E,F). Moreover, the GBA^ΔTT/ΔTT^ mutants reared on 500 nM TGZ showed a reduction in total sleep deficits that was not observed when reared on TGZ at higher doses (Figure 1G–I). PGZ at 500 nM failed to reduce daytime sleep deficits in GBA^ΔTT/ΔTT^ mutants (Figure 2A), but the GBA^ΔTT/ΔTT^ mutants reared on 1 μM PGZ treatment displayed a significant increase in daytime sleep when compared with the GBA^ΔTT/ΔTT^ mutants on control DMSO food (Figure 2B). At 2 μM, PGZ had no effect on daytime sleep (Figure 2C). PGZ exerted no effects on nighttime sleep at any dose (Figure 2D–F). There were no changes in total sleep in the GBA^ΔTT/ΔTT^ mutants that were reared on 500 nM and 2 μM PGZ drug food, but the GBA^ΔTT/ΔTT^ mutants reared on 1 µM PGZ exhibited a significant increase in total sleep, as compared to the GBA^ΔTT/ΔTT^ mutants on control DMSO (Figure 2G–I). Therefore, we concluded that pioglitazone exerts its positive effects in our neuronopathic GD fly model at an optimal dose of 1 μM. Diurnal hourly binned sleep profiles showed a reduction in daytime sleep deficits in GBA1^ΔTT/ΔTT^ mutants treated with 1 μM PGZ, with no effects on control GBA1^+/+^ flies (Appendix B Figure A2A). 

#### 2.1.3. Locomotor Deficits Reduced by 1 μM Pioglitazone

GD-associated PD is almost identical to sporadic PD. Patients with GD-associated PD exhibit younger age of onset, higher frequency of cognitive decline, slow movement, and a greater frequency of muscle rigidity. Given previous studies demonstrating the neuroprotective and mobility effects of PGZ at 1 μM in a Drosophila model of amyotrophic lateral sclerosis [37], we also analyzed the effects of 1 μM PGZ treatment on locomotor activity in the GBA^ΔTT/ΔTT^ mutant flies compared to the control GBA1^+/+^ flies. We observed that GBA^ΔTT/ΔTT^ mutant flies on control DMSO food exhibited a higher frequency of midline crossing in the daytime and nighttime (Figure 3A,B). By contrast, the GBA^ΔTT/ΔTT^ mutant flies that received 1 μM PGZ treatment did not show any significant differences between the frequency of daytime midline crossings when compared to control GBA1^+/+^ flies reared on DMSO and 1 μM PGZ (Figure 3A). The diurnal profiles of activity were similar between the groups (Figure 3C,D), likely reflecting the diurnal rest-activity cycles (Appendix B Figure A2A,B). To distinguish the effects of mobility from arousal-related measures in GBA^ΔTT/ΔTT^ mutant flies, we tested for reflex locomotor activity using the negative geotaxis assay [38,39]. The negative geotaxis assay measures climbing ability and has been used as a reliable index for mobility defects in flies [38]. The results showed that the GBA^ΔTT/ΔTT^ mutant flies on control DMSO food exhibited a decline in their ability to climb (Figure 3E) compared to the control flies on DMSO and the flies treated with 1 μM PGZ, which reduced their climbing defects (Figure 3E). 

#### 2.1.4. GBA Deficiency Leads to Lysosomal-Autophagic Deficits Reduced by 1 μM Pioglitazone

Autophagic dysfunction has been implicated in the pathophysiology underlying PD [40,41,42,43,44]. Neuronopathic fly GD models accumulate proteins normally degraded through the lysosome-autophagy pathway [15,16]. In order to determine whether the lysosome-autophagy process in our model is affected by PGZ treatment, we measured the accumulation of the lysosome-autophagy related protein Ref(2)P, the fly homolog of p62, an indicator of autophagic flux [45]. Western blot analysis showed elevated levels of Ref(2)P protein in the GBA1^ΔTT/ΔTT^ mutant flies, compared to the control flies on DMSO and 1 μM PGZ, with reduced-to-normal levels in the 1 μM PGZ-treated GBA^ΔTT/ΔTT^ mutant flies (Figure 4A,B). 

## 3. Discussion and Conclusions

The evidence provided in this study supports the development of drugs related to the thiazolidinedione PGZ for the treatment of GD/PD. We show that changes in sleep-wake behavior were present in a neuronopathic GD fly model triggered by the deletion of the fly *GBA1* gene. Administration of PGZ during development reduced the daytime and total sleep deficits in the GBA1^ΔTT/ΔTT^ mutant flies, as well as the climbing defects. We also observed increases in activity toward the end of the day and the end of the night in the GBA1^ΔTT/ΔTT^ mutant flies compared to the controls. Rather than a general increase in activity, another interpretation is that GBA1 in mutant flies could have a clock-dependent effect and alter evening and morning anticipatory activity (Figure 3C,D). While our results corroborate previous studies in other model organisms for the effective use of glitazones as neuroprotective agents [23,24,25,26,27], the present study only considered glitazone treatment by feeding flies throughout their development and testing post-eclosion. Fly dGBA1b is ubiquitously expressed in both the larval and adult brain [13,14]; therefore, our study design supports an effect of glitazone treatment that is dependent on development. While this would suggest a developmental treatment regimen for human carriers of GBA1 mutations, we cannot rule out the possibility that glitazone treatment during adult stages may also delay disease progression. Another caveat is that our studies were restricted to males, and given the phenotypic differences between the genders, the effects of glitazone should also be determined for females. Therefore, to further evaluate the efficacy of glitazone drugs at adult stages, future studies should determine whether treating GBA1^ΔTT/ΔTT^ mutant flies with glitazone drugs following eclosion would be able to reduce neurological phenotypes and corresponding cellular and molecular changes compared with young and aged flies of both genders.

Inconsistencies between TGZ and PGZ dosage efficacy on rescuing sleep in GBA1^ΔTT/ΔTT^ mutant flies were observed in this study. We observed nighttime and total sleep reduction in the 500 nM TGZ-treated GBA1^ΔTT/ΔTT^ mutant flies compared to the GBA1^+/+^ flies. We also observed daytime sleep reduction in the GBA1^ΔTT/ΔTT^ mutant flies treated with 1 μM PGZ. This may indicate differences between drug targets in cellular and molecular pathways that differentiate daytime sleep from nighttime sleep, and/or possible differential effects of drug interactions with non-sleep (i.e., circadian) pathways. We chose to pursue PGZ in follow-up mobility studies using the GBA1^ΔTT/ΔTT^ mutant due to the improvements previously observed with an optimal dose of 1 μM PGZ [37]. Future studies determining the differences between types of glitazone drugs on sleep and circadian systems will be important for considering toxicity, dose-optimization, and treatment times over the course of the day.

Although GBA1 mutations are the greatest genetic risk factor for the development of PD, the GBA1^ΔTT/ΔTT^ mutation was not shown to cause dopaminergic neuron loss in our fly model [15], in contrast to some other established PD fly models. Therefore, future studies on other PD fly models should be used to determine the general efficacy of glitazone treatment on the dopaminergic cell loss and associated phenotypes observed in our study. While it is unclear why *GBA* mutations lead to selective neuronal pathology and ultimately PD, defects in autophagy have been implicated in the pathophysiology of neurodegenerative diseases, and an impairment in lysosomal-autophagic pathways is increasingly viewed as a major process underlying neurodegeneration. The depletion of the GBA enzyme leads to the toxic accumulation of proteins and substrates within the lysosome, compromising lysosomal degradation. GBA deficiency in our GBA1^ΔTT/ΔTT^ mutant flies led to the accumulation of Ref(2)P, a marker of autophagic flux. In our study, we observed a reduction in the accumulation of Ref(2)P following PGZ treatment in the GBA1^ΔTT/ΔTT^ mutant flies. The mechanism by which PGZ may be improving the process of autophagy is unclear, but a recent study found that PGZ protects against hypoxic injury by enhancing autophagy, as evidenced by a decrease in p62 in kidney cells [46]. These results suggest that PGZ may represent a potential compound for treating PD and that therapies that target lysosomal dysfunction, including autophagic anomalies, may be effective in treating GD-associated PD. A deeper understanding of how neurodegenerative diseases such as GD-associated PD progress will offer insights into the development of novel drug targets for developing disease-modifying therapies for the effective treatment of GD-PD, such as PGZ. This study provided the opportunity to screen two different currently approved diabetic drugs for neuroprotective effects in vivo. Future efforts will be aimed at investigating the effects of PGZ on longevity, long-term memory, lipid handling, mitochondrial dysfunction, and neurodegeneration. 

To date, no drug has proven to be neuroprotective in PD. A study surveying ongoing clinical trials for therapeutics targeting PD showed that 39% are focused on long-term disease-modifying therapies, with the majority of trials (61%), focused on symptomatic relief [47]. Currently, the main mode of treatment is the administration of the DA precursor, levodopa [48]. Current treatments for patients with GD include enzyme replacement therapy, designed to supply the GBA enzyme in cells that are lacking the enzyme and substrate reduction therapy (SRT), which reduces the excess glucosylceramide substrate. There is currently no evidence that these treatments can reverse, stabilize, or slow the progression of parkinsonian neurological impairments associated with GD. As currently approved treatments for PD are merely symptomatic in nature, there is a need for effective therapies that can modify the progression of the disease. The data from this present study support the future drug development of TZD-related compounds that revert lysosomal-autophagy disrupted pathways for treatment of GD/PD.

## 4. Materials and Methods

### 4.1. Drosophila Strains

GBA1^ΔTT/ΔTT^ mutant flies (which carry a homozygous C-terminal truncation in dGBA1a and homozygous deletion in dGBA1b) and dGBA1b (GBA1^+/+^) control flies were obtained from L.J. Pallanck lab [15]. The flies were maintained on normal media containing Dimethyl sulfoxide (DMSO; Sigma, St. Louis, MO, USA), or an equal volume of 500 nM, 1 μM, or 2 μM pioglitazone (PGZ; Cayman Chemical, Ann Arbor, MI, USA), or 500 nM, 1 μM, or 2 μM troglitazone (TGZ; Cayman Chemical, Ann Arbor, MI, USA), in DMSO and allowed to lay eggs for 3 days. The larvae were reared on PGZ, TGZ, or DMSO food until adulthood.

### 4.2. Sleep/Wake Monitoring

The flies were collected following eclosion and placed on standard cornmeal agar media. Thirty-to-thirty-two 1–3 day post-eclosion male flies per genotype and drug group were individually collected under CO_2_ anesthesia and transferred into 5 mm × 65 mm polycarbonate tubes containing normal media at one end and a yarn plug at the other end. The behavioral recordings of fly sleep were collected using the *Drosophila* Activity Monitoring System (DAMS; TriKinetics Inc., Waltham, MA USA), which measures sleep on a 12 h light:12 h dark cycle. *Drosophila* activity is monitored by counting beam breaks over 5 min bins and averaged over 60 min to obtain a read-out of fly sleep/wake behavior, and sleep is defined as a 5 min period of inactivity recorded by the DAMS, as standard in the field [35,36,49]. Zeitgeber time (ZT) 0 is defined as the time of lights on, and ZT12 is defined as the time of lights off. Daytime sleep measures were from lights on to lights off (ZT0–ZT12), nighttime measures were from lights off to lights on (ZT12–ZT24), and total sleep measures were from the full 24 h period. Flies were placed in chambers/monitors for a total of 4–5 days. To allow time for the habituation of the flies to the chambers/monitors, analyses for daytime, nighttime, and total sleep were performed for the sleep recordings for days 3–4 for all the experimental groups.

### 4.3. Mobility

Thirty-to-thirty-two 1–3-day post-eclosion male flies of each genotype reared on DMSO or an equal amount of 500 nM, 1 μM, or 2 μM PGZ, or 500 nM, 1 μM, or 2 μM TGZ were individually collected under CO_2_ anesthesia and transferred into 5 mm × 65 mm polycarbonate tubes containing normal media at one end and a yarn plug at the other end. DAMS measured counts (number of beam splits) and count per minute wake for the same group of flies over a two-day period.

### 4.4. Negative Geotaxis Assay

We used 1–3 days post eclosion male flies. Twenty flies each of a given genotype reared on DMSO or an equal volume of 1× M PGZ were loaded into a vertical column. Five trials were conducted for each vial. Each trial consisted of tapping the vial five times on the lab bench, then placing the vial down and allowing the flies to climb for a duration of two minutes. The number of flies that climbed and crossed the 10 cm mark were counted every 10 s and averaged across the trials for each vial. The flies that fell were not counted. The data shown represent the flies that remained above the 10 cm mark for the duration of two minutes. A fresh Kim wipe was used to clean the column between groups. For each genotype, the means of three independent vials were averaged. The total n for each genotype and drug group = 60 flies.

### 4.5. Western Blots

The 1–3 day old fly heads (10 males) were homogenized in a RIPA buffer and an equal volume of 2× Laemmli buffer (4% SDS, 20% glycerol, 120 mM Tris-Cl pH 6.8, 0.02% bromophenol blue, 2% β-mercaptoethanol). All the samples were heated to 100 °C for two minutes. The samples were cooled on ice and then spun at 15,000 rpm for two minutes. The supernatant for each sample was then transferred to fresh tubes. The protein extracts were electrophoresed in 4–15% precast polyacrylamide gel (BIO-RAD, Hercules, CA, USA) and transferred onto PVDF 0.45 µm pore-size (Millipore, Burlington, MA, USA) membranes. The membranes were blocked in a LI-COR (Lincoln, NE, USA) Odyssey blocking buffer. The immunodetections were performed using the following antibodies: 1:1000 rabbit anti-Ref(2)P (Abcam, Cambridge, United Kingdom) and 1:1000 mouse anti-E7 (β-Tubulin, Developmental Studies Hybridoma Bank, University of Iowa, Iowa City, IA, USA). The secondary antibodies, goat anti-mouse and goat anti-rabbit (LI-COR, Lincoln, NE, USA), were used at 1:20,000. The signal was detected using LI-COR gel imager. The densitometry measurements of the western blot images were performed using the ImageJ software and the levels of Ref(2)P were determined by ratio comparing densitometric levels to loading controls (E7). Biological replications from two individual pooled groups of 10 heads each (per genotype) were included in the western blot analysis.

### 4.6. Statistical Analysis

All the data analyses were conducted using GraphPad (San Diego, CA, USA) Prism 8 statistical software. The statistical significance between the groups was determined by one-way ANOVA followed by Tukey’s post-hoc test.

## Figures and Tables

**Figure 1 ijms-22-12740-f001:**
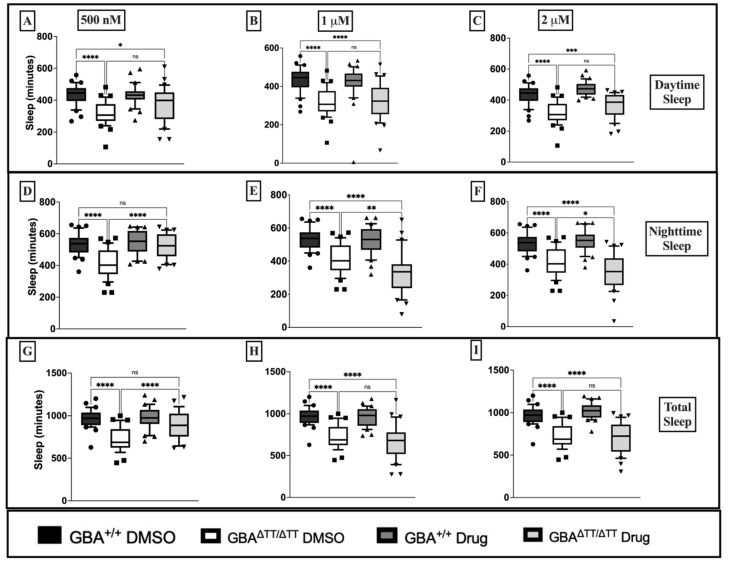
Troglitazone at 500 nM reduced nighttime and total sleep deficits in GBA1^ΔTT/ΔTT^ flies, as compared to GBA1^+/+^ flies. (**A**–**C**): Comparison of daytime sleep profile of control GBA1^+/+^ flies and GBA1^ΔTT/ΔTT^ flies on DMSO versus troglitazone drug food, assessed as minutes of sleep during the 12 h light period. (**D**–**F**): Comparison of nighttime sleep profile of control GBA1^+/+^ flies and GBA1^ΔTT/ΔTT^ flies on DMSO versus troglitazone drug food assessed as minutes of sleep during the 12 h dark period. (**G**–**I**): Comparison of total sleep profile of control GBA1^+/+^ flies and GBA1^ΔTT/ΔTT^ flies on DMSO versus troglitazone drug food assessed as minutes of sleep during the full 24 h period. *p* < 0.0001, One-Way ANOVA, and Tukey’s Post-hoc, * *p* < 0.05, ** *p* < 0.001, *** *p* = 0.0001, **** *p* < 0.0001; ns, not significant. The data are represented by box and whisker plots. Boxes represent the upper and lower quartile (25th to 75th percentile); whiskers from the 5th percentile to the 95th percentile; and the line in the box represents the median value of the data. Individual points outside of whiskers represent data outside of the 5th and 95th percentile. *n* = 30–32 flies for each genotype/drug group.

**Figure 2 ijms-22-12740-f002:**
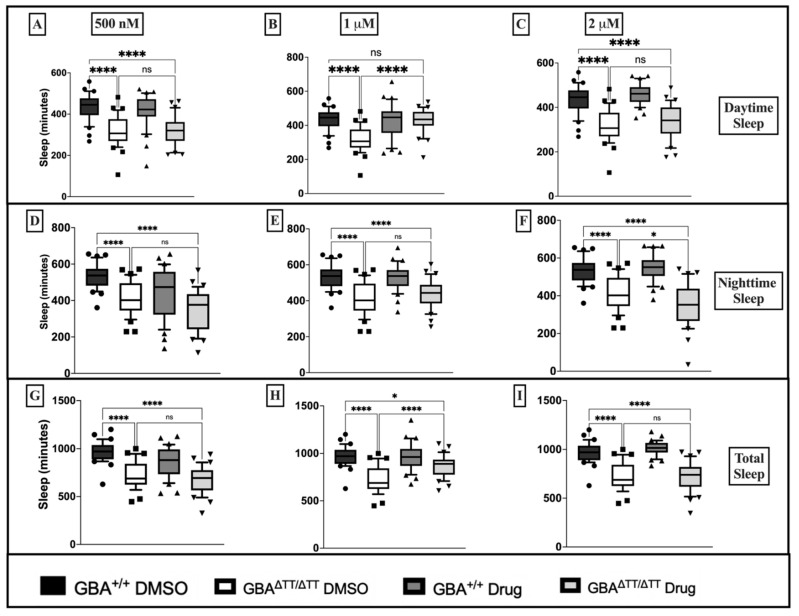
Pioglitazone at 1 μM reduces daytime sleep deficits in GBA1^ΔTT/ΔTT^ flies, as compared to GBA1^+/+^ flies. (**A**–**C**): Comparison of daytime sleep profile of control GBA1^+/+^ flies and mutant GBA1^ΔTT/ΔTT^ flies on DMSO versus pioglitazone drug food assessed as minutes of sleep during the 12 h light period. (**D**–**F**): Comparison of nighttime sleep profile of control GBA1^+/+^ flies and GBA1^ΔTT/ΔTT^ flies on DMSO versus pioglitazone drug food assessed as minutes of sleep during the 12 h dark period. (**G**–**I**): Comparison of total sleep profile of control GBA1^+/+^ flies and GBA1^ΔTT/ΔTT^ flies on DMSO versus pioglitazone drug food assessed as minutes of sleep during the full 24 h period. *p* < 0.0001, One-Way ANOVA, and Tukey’s Post-hoc, * *p* < 0.05, **** *p* < 0.0001; ns, not significant. The data are represented by box and whisker plots. Boxes represent the upper and lower quartile (25th to 75th percentile); whiskers from the 5th percentile to the 95th percentile; and the line in the box represents the median value of the data. Individual points outside of whiskers represent data outside of the 5th and 95th percentile. *n* = 30–32 flies for each genotype/drug group.

**Figure 3 ijms-22-12740-f003:**
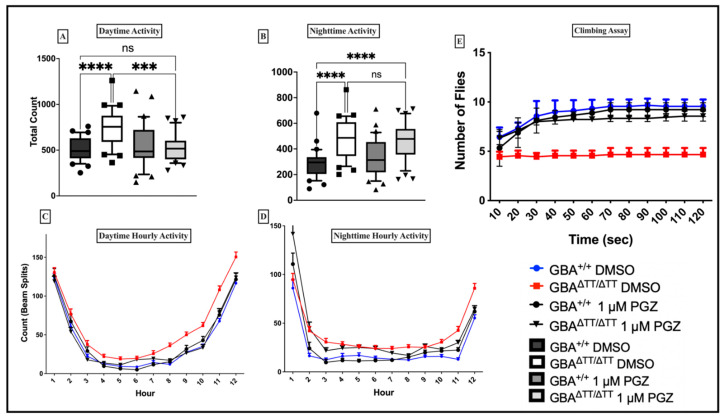
Pioglitazone reduces locomotor deficits and climbing defects in GBA^ΔTT/ΔTT^ flies at the optimal 1 µM dose. (**A**): Total number of beam splits during the 12 h light cycle in GBA^ΔTT/ΔTT^ flies is reduced with 1 µM PGZ treatment. (**B**): Total number of beam splits during the 1 h dark cycle in GBA^ΔTT/ΔTT^ flies is not reduced with 1 µM PGZ treatment. (**C**,**D**): Average counts/hour during the 12 h light period and the 12 h dark period. (**E**): GBA1^ΔTT/ΔTT^ flies showed a reduction in climbing ability that is reduced with 1 µM PGZ treatment (*p* < 0.0001; *n* = 20 flies per genotype, 3 cohorts and 5 repeats each) *p* < 0.0001, One-Way ANOVA and Tukey’s Post-hoc, *** *p* = 0.0002, **** *p* < 0.0001; ns, not significant. The data are represented by box and whisker plots in (**A**,**B**). Boxes represent the upper and lower quartile (25th to 75th percentile); whiskers from the 5th percentile to the 95th percentile; and the line in the box represents the median value of the data. Individual points outside of whiskers represent data outside of the 5th and 95th percentile. *n* = 30–32 flies for each genotype/drug group. (**C**–**E**), error bars = SD.

**Figure 4 ijms-22-12740-f004:**
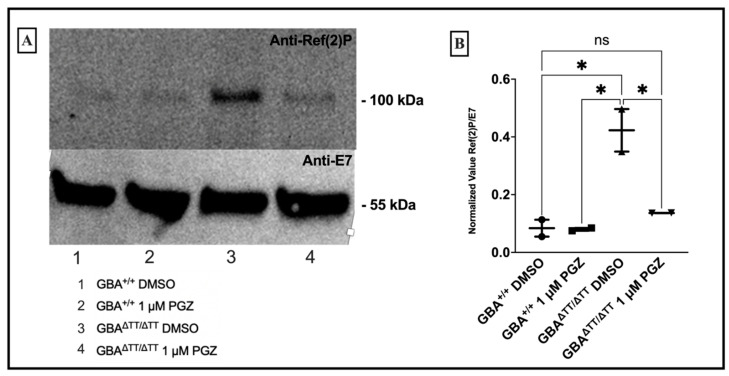
Pioglitazone returns lysosomal-autophagy signaling back to baseline levels in GBA^ΔTT/ΔTT^ flies. (**A**): Western blot analysis demonstrated increased levels of Ref(2)P in GBA^ΔTT/ΔTT^ flies. (**B**): Densitometric analysis revealed significant differences between the levels of Ref(2)P in GBA^ΔTT/ΔTT^ flies reared on DMSO compared to GBA^ΔTT/ΔTT^ reared on 1 µM PGZ and GBA1^+/+^ flies reared on DMSO and PGZ. * *p* < 0.05, One-Way ANOVA and Tukey’s post-hoc test. Error bars = SD.

## Data Availability

Not applicable.

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
