# Peer review of "Glitazone Treatment Rescues Phenotypic Deficits in a Fly Model of Gaucher/Parkinson’s Disease"

_ijms, 2021, doi:10.3390/ijms222312740_

Round 1

Reviewer 1 Report

In this manuscript the authors describe the pharmacological rescue the PD phenotype of a Drosophila GBA1 mutant. PGZ was found to be effective in restoring the sleep deficits, locomotor impairments, and climbing defects in GBA1 mutant flies. Notably, PGZ treatment also restores lysosomal-autophagy pathways, suggesting its potential use in the treatment of Gaucher Disease and Parkinson’s Disease. While the authors do not provide an explanation of the molecular mechanism of action, this work may be important for the development of new drugs to treat GD/PD.

The manuscript is overall well-written, and I have few comments and questions, which are reported below.

paragraph 4.4 Have you also tested females? 

line 296 "Flies that fell were not counted" Do you mean that those flies that passed the 10 cm line and then fell down were not counted twice? Or were they definitively excluded from the counting definitively? 

I believe the bar charts in the manuscript should be replaced. In my mind, these types of figures are no longer appropriate because they don't show the actual data. Please see this paper: 

http://journals.plos.org/plosbiology/article?id=10.1371/journal.pbio.1002128 

and adjust figures appropriately, showing the actual data.

Figure 3. In the legend (and in the methods section) it is reported that 20 flies/genotype were used. It sounds strange that in panel E there are less that 10 flies that cross the 10 cm mark (as stated in the methods section), even in the wild type strain. Please explain.

Original images of the Western blots should be provided as supplementary files.

Materials and Methods. A more detailed description of the fly strain used should be reported (e.g. genotype, and how it was obtained)

Author Response

We wish to thank the reviewer for the comments and suggestions – we feel our manuscript has been improved.

Reviewer #1 Comments and Suggestions for Authors:

paragraph 4.4 Have you also tested females?

To address this, we now include the statement in the discussion: “Another caveat is our studies were restricted to males, and given phenotypic differences between genders, the effects of glitazone should also be determined for females.”

line 296 "Flies that fell were not counted" Do you mean that those flies that passed the 10 cm line and then fell down were not counted twice? Or were they definitively excluded from the counting definitively?

We have included the following statement in methods for clarification “The recordings were done over a duration of two minutes and the final data shown represents flies that remained above the 10cm mark for the duration of the 2 minutes.”

I believe the bar charts in the manuscript should be replaced. In my mind, these types of figures are no longer appropriate because they don't show the actual data. Please see this paper:

http://journals.plos.org/plosbiology/article?id=10.1371/journal.pbio.1002128

and adjust figures appropriately, showing the actual data.

We have adjusted the figures appropriately, as suggested.

Figure 3. In the legend (and in the methods section) it is reported that 20 flies/genotype were used. It sounds strange that in panel E there are less that 10 flies that cross the 10 cm mark (as stated in the methods section), even in the wild type strain. Please explain.

We have included the following statement in methods for clarification “The recordings were done over a duration of two minutes and the final data shown represents flies that remained above the 10cm mark for the duration of the two minutes.”

Original images of the Western blots should be provided as supplementary files.

Original western blot images were added to supplemental files

Materials and Methods. A more detailed description of the fly strain used should be reported (e.g. genotype, and how it was obtained) We now include the edited statement in materials and methods for clarification: “GBA1ΔTT/ ΔTT mutant flies (which carry a homozygous C-terminal truncation in dGBA1a and homozygous deletion in dGBA1b) and dGBA1b (GBA1+/+) control flies were obtained from L.J. Pallanck lab [15].”

Reviewer 2 Report

Insects are not animals from a zoological point of view. Still, there
might be some interesting inputs from insect studies for treatment of
human/animal disease, which is in line with recent publication in IJMS
about insects and obesity. The main concept is not much to look at the
disease symptoms or neurological disorders at the human level, where
the basic mechanism underlying Gaucher/Parkinson is rather unknown,
but to study genes or mutational genes that are present not only in
insects, but also in human where they are responsible or partly
responsible of the neurological disease. If these genes are lethal
when mutated in insects remain to be proved, but in any case, the
analogy (or homology) of human/insect glucocerebrosidase systems is
certainly of interest to check for regulatory pathways and drug
effects on the gene mediating disease (in human). Of course, insects
are hardly obese and their fat metabolic pathways totally different,
but still they are recognized as very promising “medical” models (see
IJMS). This paper on fly and Parkinson is the continuity of this
insect-human homology to develop new medicinal strategies. Here, while
Introduction is very clear cut 1) clinics of Parkinson in human, 2)
genetics in fly, and 3) description of thiazolidinedione chemical used
for diabetis (in human, not in fly), the authors propose to check for
the effects of thiazolidinedione in flies, analyzing sleep, locomotion
and climbing (flying?) behavior together with some expression data of
insect/human homologs genes. There are enough numerous figures and
figure data in this work, and I have only minor questions that the
authors may consider or not for publication:
1- Why to chose males and exclude females from the analysis?
2- Are the circadian clocks (oscillator mechanisms with a periodicity
close to 24h) that control daily activity rhythm differing between
males and females? Then don't we expect some differences between flies
and human? If the results described here were dependent on the male
physiology, would it be restricted to treatment for men?
3- 1-3 days males were used for the study. Is D1 male trained the same
for circadian regulation of sleep and other behaviors than D3 males?
4- In flies, presence of the circadian clock machinery has been shown
in the brain and peripheral organs, in which organs the genes under
consideration were depleted in these mutant strains?
5- It is not mentioned in the material and methods how the “climbing”
vial was treated. It is important information because flies are
attracted by ethanol for instance and repulsed by other solvents.
6- Line 301: Bromphenol? Maybe bromophenol. Also in Western Blots,
what is the resolution of 4-15% SDS gels. Is it the mini-gel system?
What is the size and concentration of the protein to be blotted? This
is important because the efficiency of the blotting will depend on the
membrane (PVDF? Which company? Which pores?), and on the abundance of the protein in the tissue extract. Were ten heads (brain, epidermis +
antennae?) really enough to extract Ref2P? How was the protein
identity assessed? Was the blotted band subject to microsequencing?
Were the molecular weight markers visualized on the gel and the blot?
7- Figure 4 should show the position of molecular weight markers.
8- How was the densitometric value determined? Is it reliable without
loading control? What is the protein concentration in 1-2-3-4 lines?
How much protein was extracted from ten fly heads? SDS-gel should be
shown on Figure 4.
9- Figure 1 should be enlarged and include subjective sleep,
especially for a diurnal insect such as fly.
10- At what time during the photophase were the flies treated by
troglitazone? The effects are not drastic on Figure 1, including for
500 nM concentration. It would be more interesting to check for the
effects of doses< 500 nM to show or confirm the effect on the minutes
of sleep. Was the effect of Drug confirmed by recordings of
electrophysiological responses in the brain? In the antennae?
11- Same remark for Figure 2. Different doses? Different effects? How
to explain this discrepancy between the differential effects of
500nM-1uM between figure 1 and figure 2?
12- Figure 3: I am surprised to see the same activity rate between
daytime and nighttime. Flies are diurnal insects and their high
activity in daylight is genetically inherited. There maybe some age
variations in the activity during scotophase depending on the age?
There is rather high variability in the results (see standard
deviation: about 100 minutes on Figure 2, about 100 insects on Fig.3)
How about the females?
13- Figure 3CD: I see no differences between the mutants and the
treated groups for the activity measured every hour. Flies are active
during the day, at least to find food or mate, or simply being
attracted by light or any alcoholic substance but this is not the case
here. They are active apparently at the night-light switch, and this
is observed in the four groups tested. Very intriguingly, the two
curves do not match. After 12 h lights, the group 1 (red) is very
active, but after only 1 h dark the group1 is much less active, while
the mutants (+drug) are more active. Does it mean that the light-off
signal is perceived differently by mutants and mutants + drug? Then,
how to explain that there are no differences when the light is switch
on?
14- Maybe more counts are required here to validate a statistically
correct approach.
15- Figure 3E showing no climbing in mutants might be correct, what is
the effect of 50nM treatment?
16- Molecular data, i.e. gene/RNA expression analysis, qRT-PCR and/or
one step RT-PCR are required to validate Figure 4, comparison to
sleep/locomotion/climbing genes, discussion and conclusion

Author Response

We wish to thank the reviewer for the comments and suggestions – we feel our manuscript has been improved.

Reviewer #2 Comments and Suggestions for Authors:

1- Why to chose males and exclude females from the analysis?

2- Are the circadian clocks (oscillator mechanisms with a periodicity

close to 24h) that control daily activity rhythm differing between

males and females? Then don't we expect some differences between flies

and human? If the results described here were dependent on the male

physiology, would it be restricted to treatment for men?

To address 1and2, we now include the statement in the discussion: “Another caveat is our studies were restricted to males, and given phenotypic differences between genders, the effects of glitazone should also be determined for females.  There-fore, to further evaluate the efficacy of glitazone drugs at adult stages, future studies should determine whether treating GBA1ΔTT/ /ΔTT mutant flies with glitazone drugs following eclosion would be able to rescue neurological phenotypes and corresponding cel-lular and molecular changes compared between young and aged flies of both genders.”

3- 1-3 days males were used for the study. Is D1 male trained the same

for circadian regulation of sleep and other behaviors than D3 males?

Yes, taking 1-3 day old males is a standard in the field.

4- In flies, presence of the circadian clock machinery has been shown

in the brain and peripheral organs, in which organs the genes under

consideration were depleted in these mutant strains?

These are whole organism (genomic) mutations

5- It is not mentioned in the material and methods how the “climbing”

vial was treated. It is important information because flies are

attracted by ethanol for instance and repulsed by other solvents.

We now include the statement “A fresh Kim wipe was used to clean the column between groups.”

6- Line 301: Bromphenol? Maybe bromophenol. Also in Western Blots,

what is the resolution of 4-15% SDS gels. Is it the mini-gel system?

What is the size and concentration of the protein to be blotted? This

is important because the efficiency of the blotting will depend on the

membrane (PVDF? Which company? Which pores?), and on the abundance of the protein in the tissue extract. Were ten heads (brain, epidermis +

antennae?) really enough to extract Ref2P? How was the protein

identity assessed? Was the blotted band subject to microsequencing?

Were the molecular weight markers visualized on the gel and the blot?

Bromophenol spelling has been corrected.  We have now included the size of protein on the figures.  We reference the Ref2p source from Abcam and now included “PVDF 0.45µm pore-size (Millipore)” in the methods section.

7- Figure 4 should show the position of molecular weight markers.

We have now included the size of protein on the figures and the markers can be seen on the whole blot (unedited) images in supplemental data.

8- How was the densitometric value determined? Is it reliable without

loading control? What is the protein concentration in 1-2-3-4 lines?

How much protein was extracted from ten fly heads? SDS-gel should be

shown on Figure 4.

A copy of the whole gel was submitted supplemental data, densitometry was performed in ImageJ and we now include in the methods “and levels of Ref(2)P were determined by ratio comparing densitometric levels to loading controls (E7).”

9- Figure 1 should be enlarged and include subjective sleep,

especially for a diurnal insect such as fly.

We have changed the figures in response to review for alternate representation.

10- At what time during the photophase were the flies treated by

troglitazone? The effects are not drastic on Figure 1, including for

500 nM concentration. It would be more interesting to check for the

effects of doses< 500 nM to show or confirm the effect on the minutes

of sleep. Was the effect of Drug confirmed by recordings of

electrophysiological responses in the brain? In the antennae?

11- Same remark for Figure 2. Different doses? Different effects? How

to explain this discrepancy between the differential effects of

500nM-1uM between figure 1 and figure 2?

10&11 Drug treatment (as indicated in materials and methods) was throughout development.

12- Figure 3: I am surprised to see the same activity rate between

daytime and nighttime. Flies are diurnal insects and their high

activity in daylight is genetically inherited. There maybe some age

variations in the activity during scotophase depending on the age?

There is rather high variability in the results (see standard

deviation: about 100 minutes on Figure 2, about 100 insects on Fig.3)

How about the females?

13- Figure 3CD: I see no differences between the mutants and the

treated groups for the activity measured every hour. Flies are active

during the day, at least to find food or mate, or simply being

attracted by light or any alcoholic substance but this is not the case

here. They are active apparently at the night-light switch, and this

is observed in the four groups tested. Very intriguingly, the two

curves do not match. After 12 h lights, the group 1 (red) is very

active, but after only 1 h dark the group1 is much less active, while

the mutants (+drug) are more active. Does it mean that the light-off

signal is perceived differently by mutants and mutants + drug? Then,

how to explain that there are no differences when the light is switch

on?

12&13

Flies are crepuscular (morning and evening active) with a mid-day siesta; this is typically more pronounced in males; we mention the statement to address gender in the discussion: “Another caveat is our studies were restricted to males, and given phenotypic differences between genders, the effects of glitazone should also be determined for females.  There-fore, to further evaluate the efficacy of glitazone drugs at adult stages, future studies should determine whether treating GBA1ΔTT/ /ΔTT mutant flies with glitazone drugs fol-lowing eclosion would be able to rescue neurological phenotypes and corresponding cellular and molecular changes compared between young and aged flies of both genders.”

14- Maybe more counts are required here to validate a statistically

correct approach.

15- Figure 3E showing no climbing in mutants might be correct, what is

the effect of 50nM treatment?

We have the following statement in the discussion “We chose to pursue PGZ in follow-up mobility studies GBA1ΔTT/ /ΔTT mutant due to the improvements previously observed with an optimal dose of 1μM PGZ [37].  Future studies determining the differences between types of glitazone drugs on sleep and circadian systems will be important for considering toxicity, dose-optimization and treatment times over the course of the day.”

[37] Joardar A, Menzl J, Podolsky TC, Manzo E, Estes PS, Ashford S, Zarnescu DC: PPAR gamma activation is neuroprotective in a Drosophila model of ALS based on TDP-43. Hum Mol Genet 2015, 24(6):1741-1754.

16- Molecular data, i.e. gene/RNA expression analysis, qRT-PCR and/or

one step RT-PCR are required to validate Figure 4, comparison to

sleep/locomotion/climbing genes, discussion and conclusion

We were only interested in measuring the functional Ref(2)P protein (not gene expression) since this represents lysosomal-authophagy signaling in the cell (RNA does not).

Round 2

Reviewer 1 Report

The manuscript has been improved according to my previous comments. 

Author Response

We have edited the manuscript according to all suggestions - we wish to thank the reviewer for their careful consideration and thoughtful comments.

This manuscript is a resubmission of an earlier submission. The following is a list of the peer review reports and author responses from that submission.

Round 1

Reviewer 1 Report

People with diabetes mellitus appear to have a higher incidence of Parkinson disease (PD). The use of antidiabetic glitazones are apparently associated with a diminished risk of PD incidence in patients with diabetes. In vertebrate animal models of PD, glitazones have also shown their potential to act as neuroprotective drugs and reducing neuroinflammation. Several Drosophila models for PD have been established, and these flies express a variety of PD-like symptoms like sleep problems, reduced longevity, progressive age dependent motor deficits and in some cases loss of dopaminergic neurons. Here, the authors used flies, which carry mutations in the Glucocerebrosidase (Gba1) gene as the cause of Gaucher disease and a major risk factor for PD in humans. Previous analysis by other labs showed that these flies have a shortened life span, progressive impairment of climbing ability a variety of cellular phenotypes including lysosomal-autophagic defects and formation of protein aggregates.

The authors now show that Gba1 mutant flies have defects in the sleep-wake behavior (reduction in day and night time sleep). In combination with other reported phenotypes (locomotor deficits, accumulation of the autophagic protein p62), they tested the effects two different glitazones (PGZ, TGZ) at different doses for rescue of these phenotypes. The analysis confirms the effectiveness of glitazones at specific doses to revert mutant phenotypes. The findings are convincingly presented and substantiate other reports in vertebrates for potential use of glitazones as neuroprotective reagents.

Having said this general positive statement, my major concern is the experimental set-up. Glitazone treatment was done during the whole development and flies were tested within the first days after eclosion. Thus the observed improvements after treatment could be solely due to rescue of developmental defects. Second, any data about rescue of age-dependent progressive defects (e.g. climbing performance) is missing. Therefore, in my opinion, glitazone treatment should start after eclosion and the corresponding assays should be performed with young and old flies to evaluate the effectiveness of glitazones against progression of disease phenotypes. This type of analysis should also be done for the Western Blot analysis, which ideally should be complemented by a second approach (e.g. Lysotracker).

Minor points

- what is the definition of sleep in this study

- “Sleep was analyzed done over 4 days and then averaged over 2 days.” The authors should indicate which of the 2 days were selected and whether all experimental groups were analyzed at the same two days.

- Looking at activity profiles in Fig. 3C, D my impression is that Gba1 mutant flies have an increased morning and evening anticipation rather than a general increase in activity. This might indicate a clock dependent function of Gba1.

- Given the rescue of daytime and nighttime sleep deficits by TGZ, I am wondering why the authors used PGZ for further experiments, which only rescued daytime sleep.

- The authors should discuss the differential effects of TGZ and PGZ on daytime and nighttime sleep

- Labeling of Fig 3C,D is too small to read.

- Please indicate how many biological replicates were used for Western blot analysis.

Reviewer 2 Report

This manuscript reports glitazone treatment rescued sleep deficits, locomotor impairments, climbing defects, and restoration of normal protein levels of P62/Ref(2)P in a fly model of Gaucher disease  (GBA1ΔTT/ ΔTT ). 

Some comments below:

Not sure why the authors call the GBA1ΔTT/ΔTT mutant fly GD model as GD/PD model.  GBA1 is a strong genetic risk factor of Parkinson Disease, however that does not make GD model a PD model.  e.g. GBA1ΔTT/ΔTT does not cause dopaminergic neuron loss, in contrast to some other established PD fly models.  Whole paper much emphasize on PD rather than Gaucher disease itself, which I find it unjustified.

In figure 1, TGZ was tested and shown 500nM was able to rescue nighttime sleep deficits.  This was not further discussed anywhere in the paper.  Why did higher concentrations (1 and 2 uM) have no effects? Are they toxic to the flies at that conc? Similar question on PGZ @ 2uM

PGZ (1uM) rescued daytime sleep and daytime activity deficits, whereas TGZ (500nM) rescued nighttime sleep deficits.  Does rescuing daytime vs nighttime deficits by PGZ or TGZ reflecting any difference on how they act?

In the abstract the authors state “Gaucher disease, a rare lysosomal storage disorder”.  This is somewhat incorrect.  Lysosomal storage disorders overall are rare, but Gaucher disease is actually the most common type of lysosomal storage disorder.

Ref #46: is it correct ref for PGZ enhancing autophagy (line 203)? no PGZ was used in that ref.